# Dynamic Correction of the Influence of Long Measuring Path Irregularity in Antenna Tests

**Elena Dobychina * and Mikhail Snastin**

Scientific and Production Center of Radio Information Metrology, Moscow Aviation Institute, 125993 Moscow, Russia; mexanizmys@ya.ru
* Correspondence: dobychina_mai@mai.ru

**Abstract:** This article investigates the influence of random microwave discontinuities on the characteristics of long transmission paths. This is most important for dynamic measuring stands, accompanied by multiple space movement of long transmission paths with their bending or twisting during the measurement process. In modern active electronically scanned arrays this issue also becomes relevant, due to increased requirements for the accuracy of beam shaping. The aim of this study is to develop a theoretical background and perform experimental verification for taking into account the effect of random microwave discontinuities on the transmission path characteristics. A method for correcting the effect of such irregularities is considered based on electrical length control by measuring the input reflection coefficient. Relations for the magnitude and phase of the path's input reflection coefficient depending on the S-parameters of a long four-terminal network terminated with mismatched load are obtained and plotted. Using theory of sensitivity, the mathematical expressions of conditions were obtained to achieve maximum accuracy of measuring the electrical length of a long microwave path. The possibility of dynamic error correction in antenna measurements with a long test path caused by random microwave irregularities along it has been experimentally proved.

**Keywords:** long transmission paths; dynamic measuring; random microwave discontinuities; dynamic error correction

## 1. Introduction

At present, various microwave systems are being developed, which require high stability of the oscillation signal parameters. Such systems include, for example, active electronically scanned arrays (AESA) [1,2], providing a narrow radiation pattern (RP). On the other hand it can be monopulse antenna arrays, which have a high value of difference pattern null slope, and therefore a high resolution. Thus, it expands the accessible amount of information about the propagation of electromagnetic waves from sources of radiation or reflection in the space around. However, these advantages are realized due to high identity of the channels output signals that is subject to various destabilizing factors.

All measurement errors arising under the influence of destabilizing factors can be grouped into systematic and random ones. Systematic errors can be partially removed by introducing some compensating correction during the initial setup, whereas random errors cannot be systematized. To take into account the influence of random errors on the output signal variation of long microwave transmission lines, it is necessary to create special methods and algorithms for their determination and compensation.

Radio engineering systems, which include antennas and antenna arrays, require increased accuracy of test procedures [3]. Ensuring high-precision measurements of the antenna technology parameters allows one to increase the development efficiency of the complex devices, potentially improve the technical performance of systems used and verify the characteristics of off-the-shelf products [4–7].

Antenna measurements are carried out in the near [8] or far field [9] regions using specialized test facilities such as anechoic or reverberation chambers [10], free-space test

ranges [11] or other equipment types. At the same time, not only the measured data themselves are of interest, but also issues related to the estimation of the measurement error [12], as well as the measurement reproducibility [13]. It is often necessary to provide the relative movement between the measuring antenna and the antenna under test (AUT) during a measurement process, so measurements are not static in the general case. An example is AUT rotation in the azimuthal plane when measuring cross-sections of RP [14] or near-field scanning to obtain the amplitude-phase distribution with the probe [12]. The effect of changes in test cable characteristics due to twisting or bending is usually not taken into account in such measurements.

The aims of the work were to systematize the theoretical aspects of taking into account the influence of random factors on the characteristics of long measurement paths, develop an error correction mechanism and verify the proposed calibration method.

As a result, relations were obtained indicating the need to correct random errors at the output of long microwave transmission lines found in both AESA feeding systems and measuring systems.

## 2. Mathematical Model of a Long Microwave Path

When investigating the output signal of various transmission systems, it is necessary to take into account all the factors affecting their stability over time. In general, errors occur as a result of combining two types of processes, namely slow and fast. A slow process is associated with a variation in electrical length of the signal path due to both environmental conditions and the electrical parameters drift of its component parts. In turn, fast process is associated with thermal, shot and flicker fluctuations.

Let us compare errors at the output of a long microwave path based on the theory of network synchronization before and after applying the correction [15]. To do that, we write output signal of the signal source as:

$$\dot{S}(t) = A(t) sin\, \dot{\Phi}(t),$$

$$A(t) = A + \delta A(t),$$

where $\delta A(t)$ is an amplitude variation of the signal source output, relative to some fixed value $A$ (amplitude fluctuations); $\dot{\Phi}(t)$ is a phase at the signal source output. Since the phase at the output of the signal source changes with time, we will denote the function describing the behavior of the phase in time by the term "phase process".

Next, we write the function of the instantaneous angular frequency of the signal source [16].

$$\dot{\omega}_G(t) = \omega_0 + \sum_{k=0}^{M-1} \frac{Q(k)}{k!} t^k + \dot{\xi}(t), \tag{1}$$

where $\omega_0$ is the nominal value of the natural angular frequency of the signal source; $Q(0)$ is a random value with a zero mean that expresses the initial frequency error. This error occurs due to uncertainty in the initial setting of the source's natural frequencies (setting error); $Q(k)$, $(k = I, \ldots, M - I)$ is a set of time-independent random variables that simulate frequency drifts of $k$-th order (long-term instability); $\dot{\xi}(t)$ is a stationary random process with zero mean, describing short-term instabilities of the signal source.

The phase process of the signal source is found as a result of integration (1) on interval $[0, t]$:

$$\dot{\Phi}_G(t) = \Phi(0) + \omega_0 t + \sum_{k=1}^{M} \frac{Q(k-1)}{k!} t^k + \left[ \dot{\xi}(t) - \dot{\xi}(0) \right], \tag{2}$$

where $\left[ \dot{\xi}(t) - \dot{\xi}(0) \right]$ is generally a non-stationary stochastic process that describes short-term instabilities of the reference signal source (phase noise).

Consider two types of long signal paths—without error correction and with full error correction. The first type includes, for example, passive systems for signal routing in the

form of a series-parallel combination of power dividers and transmission lines [17], as well as test cables between the measuring equipment and AUT during antenna tests [18]. Based on (2), let us write the phase process at the output of any $v$-th channel of such a system:

$$\dot{\Phi}_v(t) = \Phi(0) + \omega_0 t + \sum_{k=1}^{M} \frac{Q(k-1)}{k!} t^k + \left[ \dot{\xi}(t) - \dot{\xi}(0) \right] + \beta l_v(t) + N_{kv}(t), \tag{3}$$

where $\beta l_v(t) = L_v(t)$ is electrical length of a given channel; $N_{kv}$—additive thermal noise of $v$-th channel.

The mathematical model of the passive signal routing unit of the source signal without phase stabilization can be written in matrix representation as:

$$\mathbf{\Phi}_p(t) = \mathbf{\Phi}_G(t)\gamma + \mathbf{L}(t) + \mathbf{N}_k(t), \tag{4}$$

where $\mathbf{\Phi}_p(t) = [\Phi_1(t), \Phi_2(t), \dots \Phi_n(t)]^T$ is a column vector of the output phase process; $n$ is the total number of output channels of the system; $\mathbf{\Phi}_G(t)$ is the phase process at the output, determined by (2); $\gamma = [\underbrace{I, I, \dots I}_{n}]^T$ is identity column vector,

$$\mathbf{L}(t) = [L_1(t), L_2(t), \dots L_n(t)]^T, \ \mathbf{N}_k(t) = [N_{k1}(t), N_{k2}(t), \dots N_{kn}(t)]^T.$$

Expression (4) shows that the phase process at each channel output of a passive long signal path without phase stabilization is determined by the phase process of the signal source, the channel electrical length and its equivalent thermal noise.

The second class of signal paths includes transmission systems with a driving oscillator, the parameters of which are stabilized either by correction or by auto-tuning for each channel separately [19]. The phase process at the output of the $v$-th channel of such a system in generalized form is written as:

$$\dot{\Phi}_v(t) = \dot{\Phi}_G(t) + \sum_{j=1}^{k} a_{vj} \left\{ K_{vj} Y_{vj}(p) \left[ g_{vj} \left[ \dot{\Phi}_{vj}(t) \right] + \varepsilon_{vj}(t) \right] \right\}, \tag{5}$$

where $k$ is the number of steps (hierarchy levels) for distributed transmission systems; $K_{vj}$ represents gain of the correction loop for $v$-th channel at $j$-th step; $Y_{vj}$ denotes the transfer function of correction loop; $g_{vj}[\dots]$ is phase discriminator frequency response; $\alpha_{vj}$ stands for weight coefficients of adder (in the self-correction system, $\alpha_{vj} = I$); $\varepsilon_{vj}$ is the equivalent phase noise of the channel. A mathematical model of the system with a driving oscillator and a correction can be written in the form:

$$\mathbf{\Phi}_S(t) = \mathbf{\Phi}_G(t)\gamma + \mathbf{A}(t), \tag{6}$$

where $\mathbf{A}(t) = [A_1(t), A_2(t), \dots A_N(t)]^T$ is a column vector containing the diagonal of a square matrix $\mathbf{D}(t)$ of $N \times N$ size: $A_v(t) = D_{vj}(t)\big|_{v=j}$. Matrix $\mathbf{D}(t)$ is defined by the expression:

$$\mathbf{D}(t)_{N \times N} = \mathbf{Y}(p)\mathbf{G}^T(t) \tag{7}$$

$$\mathbf{G}(t) = \mathbf{B}(t) + \mathbf{N}(t),$$

$$\mathbf{B}(t) = [K_{vj} g_{vj} [\Phi_{vj}(t)]]_{N \times K} \tag{8}$$

$$\mathbf{N}(t) = [K_{vj} \varepsilon_{vj}(t)]_{N \times K} \tag{9}$$

$$\mathbf{Y}(p) = [Y_{vj}(p)]_{N \times K} \tag{10}$$

The phase process at the output of each channel of considered excitation system is determined, in particular, by the phase of the signal source, the system structure, the gain

of the correction loop, the discriminator frequency response, and the phase noise of the system, as shown through expressions (6)–(10).

Comparing mathematical models of two classes of excitation systems (4) and (6), we can conclude that the accuracy of phase stabilization of the output signal in first class systems is determined by difference in electrical lengths of the channels, the phase noise of the source, as well as channels themselves. Thus, total phase error in such a system can significantly exceed tens of degrees due to unregulated changes in the path's electrical length (for example, due to changing environmental conditions or mechanical impact). In second class systems, the total error is determined by the parameters of the correction system; therefore, it is possible to minimize it using a dynamic calibration.

### 3. Analysis of the Electrical Length Control Method

Long transmission path stabilization is determined by two factors: the accuracy of the electrical length control and the total error tracking. The accuracy of phase error tracking is determined by the parameters of stabilization system. The accuracy of the electrical length control is determined by the chosen measurement method.

A number of works have been devoted to methods for measuring the electrical length of long paths [20–23]. The viability of using a particular method depends on the specific situation. In our work, a measurement method using load reflections was investigated. The method of unmodulated reflection from the load has found application mainly in the impedance and reflection coefficient measurements [22], as well as in power supply systems of linear accelerators [19]. It is proposed to use this method for measuring the electrical length of a long microwave path as follows.

When oscillations are excited in the lossless line terminated with a mismatched load, a part of the incident wave, reflected from the load propagates in the opposite direction. Due to the interference of the incident and reflected waves, the mixed standing and travelling wave mode is established in the line. When changing the line length, the waves shift along it, by an amount equal to the electrical length deviation from the nominal value. By registering the standing wave shifting, for example, on the displacement of the wave nodes, it is possible to control the electrical length of the microwave path.

The accuracy of electrical length control with the presented method is influenced by several factors: reflections from microwave discontinuities within the path, the cable attenuation, the load reflection, as well as the signal source reflection. With a theory of sensitivity, we will estimate the degree of influence of these factors on the oscillation signal in case of the electrical length control [24]. Sensitivity analysis is associated with the study of the effect of parameters variation on the behavior of dynamic systems. The goal is to find the sensitivity matrix:

$$\boldsymbol{U}(p) = \left[\frac{\partial \boldsymbol{x}}{\partial \boldsymbol{p}}\right] = \left[\frac{\partial x_n}{\partial p_m}\right], \tag{11}$$

where $\boldsymbol{x}$ is the $N$-dimensional vector of the system state; $\boldsymbol{p}$ is the $M$-dimensional vector containing system parameters.

In practice, the sensitivity matrix is most often determined with the mathematical model of the system [25]. To design and analyze the model, we will use the well-known theory of directed graphs [26,27]. A measuring system for determining the electrical length of a long transmission line using the unmodulated reflection method can be considered as a cascade connection of several blocks. The first of them is the two-terminal device, which represents a signal source, then there is the four-terminal device, which stands for a transmission path with microwave discontinuities and attenuation, and finally the two-terminal device, which acts as the load (Figure 1).

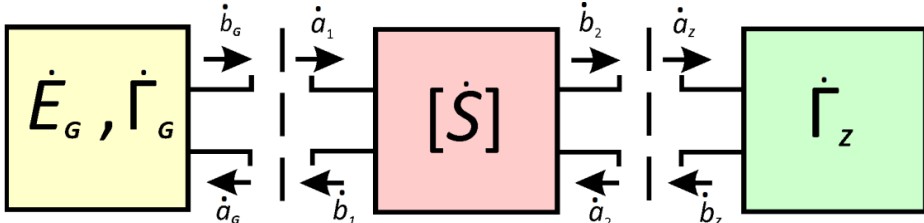

**Figure 1.** Schematic diagram of measuring system for electrical length of the transmission path.

Each of the n-terminal networks of the system is assumed to be linear and described by the corresponding equation:

$$
\begin{array}{ll}
\dot{b}_G = \dot{E}_G + \dot{a}_G\,\dot{\Gamma}_G & \text{– signal source} \\[4pt]
\begin{pmatrix} \dot{b}_1 \\ \dot{b}_2 \end{pmatrix} = \begin{bmatrix} \dot{S}_{11} & \dot{S}_{12} \\ \dot{S}_{21} & \dot{S}_{22} \end{bmatrix} \begin{pmatrix} \dot{a}_1 \\ \dot{a}_2 \end{pmatrix} & \text{– transmission path} \\[4pt]
\dot{b}_Z = \dot{a}_Z\,\dot{\Gamma}_Z & \text{– load}
\end{array}
\tag{12}
$$

where $\dot{a}_1, \dot{a}_2, \dot{a}_G, \dot{a}_Z$ are incident waves at the input and output of the transmission path, the source output and load input, respectively; $\dot{b}_1, \dot{b}_2, \dot{b}_G, \dot{b}_Z$ are corresponding reflected waves; $\dot{E}_G$—the source voltage value; $\dot{\Gamma}_G, \dot{\Gamma}_Z$—complex reflection coefficients of the source and load, respectively; $\dot{S}_{11}, \dot{S}_{22}, \dot{S}_{12}, \dot{S}_{21}$—scattering parameters of the four-terminal network of a microwave transmission path.

To analyze the measuring accuracy of the system parameters, it is necessary to obtain an analytical expression for the reflection coefficient at the input of microwave path. When combining element blocks in pairs in accordance with (12), the same wave is incident for one of them and reflected for the other, i.e.,

$$
\dot{a}_G = \dot{b}_1,\ \ \dot{b}_G = \dot{a}_1,\ \ \dot{a}_2 = \dot{b}_Z,\ \ \dot{b}_2 = \dot{a}_Z
$$

Since one of the connected nodes for each pair is a source, a direct connection of the considered directed graphs is acceptable. Figure 2a shows a total directed graph of the circuit combined. Figure 2b shows the result of simplifying the graph obtained.

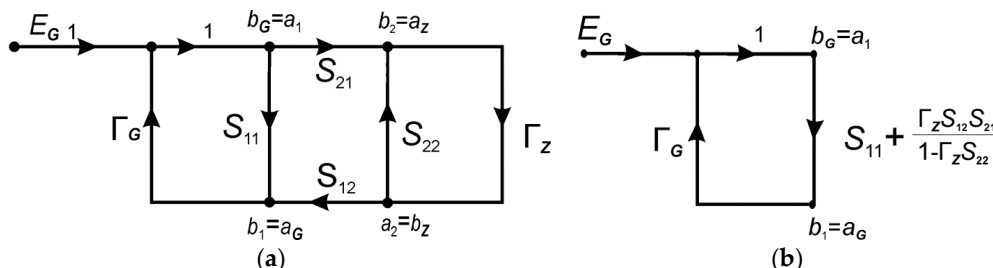

**Figure 2.** Directed graph of the electrical length control system: (**a**) total; (**b**) simplified.

It can be used to write an expression for the input reflection coefficient of a four-terminal network:

$$
\dot{\Gamma}_{in} = \frac{\dot{a}_G}{\dot{E}_G} = \frac{\dot{S}_{11}\left(1 - \dot{\Gamma}_Z\dot{S}_{22}\right) + \dot{\Gamma}_Z\dot{S}_{12}\dot{S}_{21}}{1 - \dot{\Gamma}_Z\dot{S}_{22} - \dot{S}_{11}\,\dot{\Gamma}_G + \dot{\Gamma}_G\,\dot{\Gamma}_Z\dot{S}_{11}\dot{S}_{22} - \dot{\Gamma}_G\,\dot{\Gamma}_Z\dot{S}_{12}\dot{S}_{21}}
\tag{13}
$$

To obtain the magnitude and phase of the input reflection coefficient of the path, it is useful to represent (13) in trigonometric form:

$$\dot{\Gamma}_{in} = \Gamma_{in}\cos\varphi_{in} + j\Gamma_{in}\sin\varphi_{in}$$
$$\dot{S}_{11} = S_{11}\cos\varphi_{11} + jS_{11}\sin\varphi_{11}$$
$$\dot{S}_{12} = S_{12}\cos\varphi_{12} + jS_{12}\sin\varphi_{12}$$
$$\dot{S}_{21} = S_{21}\cos\varphi_{21} + jS_{21}\sin\varphi_{21}$$
$$\dot{S}_{22} = S_{22}\cos\varphi_{22} + jS_{22}\sin\varphi_{22}$$
$$\dot{\Gamma}_{G} = \Gamma_{G}\cos\varphi_{G} + j\Gamma_{G}\sin\varphi_{G}$$
$$\dot{\Gamma}_{Z} = \Gamma_{Z}\cos\varphi_{Z} + j\Gamma_{Z}\sin\varphi_{Z}$$

The elements of the sensitivity matrix of the investigated system (11) are determined from (13). In this case, $x = [\Gamma_{in}, \varphi_{in}]$ is the vector describing the state of the system $N = 2$; $p = [S_{11}, S_{12}, S_{21}, S_{22}, \varphi_{11}, \varphi_{12}, \varphi_{21}, \varphi_{22}, \Gamma_{G}, \varphi_{G}, \Gamma_{Z}, \varphi_{Z}]$ is the vector containing system parameters, $M = 12$.

The following characteristics shown in Figure 3a–f are particularly interesting for the study of the microwave discontinuities influence on the electrical length control of the transmission path.

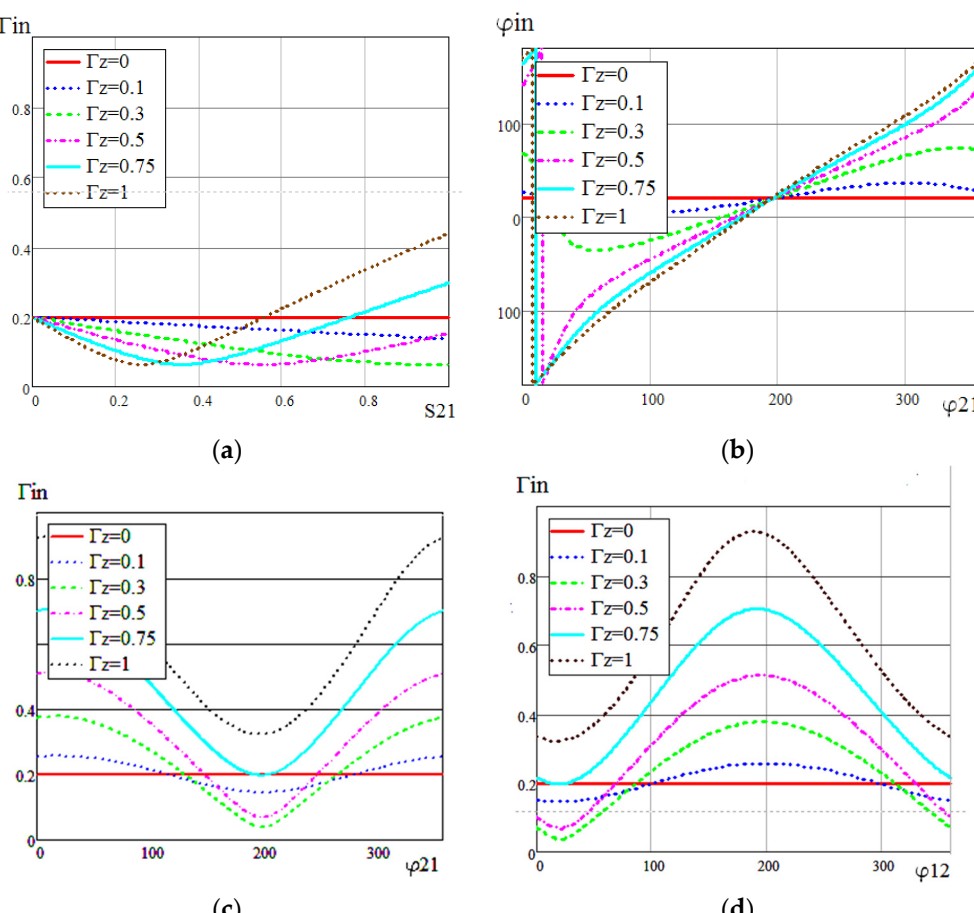

**Figure 3.** *Cont*.

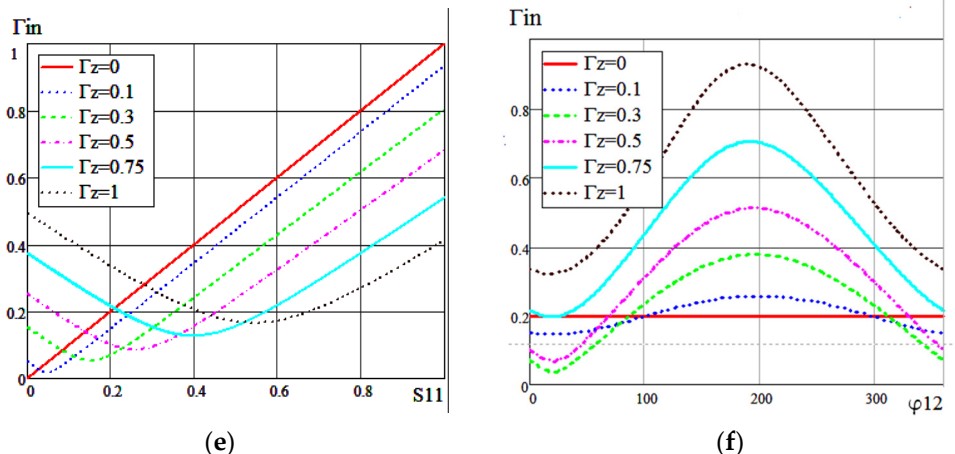

**Figure 3.** Magnitude of the input reflection coefficient of a long microwave path depending on: (**a**) $S_{21}$; (**c**) $\varphi_{21}$; (**d**) $\varphi_{12}$; (**e**) $S_{11}$ and phase depending on: (**b**) $\varphi_{21}$; (**f**) $\varphi_{11}$.

Based on the obtained results, preliminary conclusions can be drawn that are useful for the electrical length control of long microwave paths:

- to improve the accuracy of $\varphi_{21}$ control by the method of unmodulated reflection from the load, it does not need to be perfectly matched with line. Moreover, the value of the reflection coefficient from the load can reach $\Gamma_Z = 0.5 \dots 0.75$ (Figure 3a,b);

- the non-identity of the $\Gamma_{in}(\varphi_{12})$ and $\Gamma_{in}(\varphi_{21})$ characteristics of the microwave path leads to measuring errors the electrical length due to changes in $\Gamma_Z$ (Figure 3c,d);

- to reduce the influence of $S_{11}$ and $\varphi_{11}$ (the direct wave reflection from irregularities) on the measuring accuracy of the path's electrical length, it is necessary to increase $\Gamma_Z$ (Figure 3e,f).

Consider sensitivity matrix elements of the system (11), which characterize the influence of the parameters $\varphi_{11}$, $\varphi_{12}$, $\varphi_{21}$, $\varphi_{22}$, $\varphi_Z$ on the change in the phase of $\varphi_{in}$. Analysis of these elements will make it possible to mathematically write the conditions for the maximum accuracy of the electrical length control. Based on (11) we can write

$$\frac{d\varphi_{in}}{d\varphi_{11}} = \frac{S_{11}^2 + \Gamma_Z S_{11} S_{12} S_{21} \cos\theta}{S_{11}^2 + (\Gamma_Z S_{12} S_{21})^2 - \Gamma_Z^2 S_{11} S_{22} S_{12} S_{21} \cos F + 2\Gamma_Z S_{11} S_{12} S_{21} \cos\theta}; \tag{14}$$

$$\frac{d\varphi_{in}}{d\varphi_{22}} = \frac{\Gamma_Z^2 S_{22} S_{12} S_{21} [\Gamma_Z S_{12} S_{21} \cos(\varphi_Z + \varphi_{22}) + 2S_{11}\cos\theta - S_{11}\cos F]}{S_{11}^2 + \Gamma_Z S_{12} S_{21} [\Gamma_Z S_{12} S_{21} + 2S_{11}\cos\theta - 2\Gamma_Z^2 S_{22} S_{12} S_{21} \cos(\varphi_Z + \varphi_{22})]}; \tag{15}$$

$$\frac{d\varphi_{in}}{d\varphi_{12}} = \frac{d\varphi_{in}}{d\varphi_{21}} = \frac{(\Gamma_Z S_{12} S_{21})^2 + \Gamma_Z S_{11} S_{12} S_{21} \cos\theta}{S_{11}^2 + (\Gamma_Z S_{12} S_{21})^2 + 2\Gamma_Z S_{11} S_{12} S_{21} \cos\theta} + \frac{\Gamma_G \Gamma_Z S_{12} S_{21} \cos\psi}{1 - 2\Gamma_Z S_{22} \cos(\varphi_Z + \varphi_{22}) - 2\Gamma_G \Gamma_Z S_{12} S_{21} \cos\psi}; \tag{16}$$

$$\frac{d\varphi_{in}}{d\varphi_Z} = \frac{(\Gamma_Z S_{12} S_{21})^2 + S_{11} \Gamma_Z S_{12} S_{21} \cos\theta}{S_{11}^2 + (\Gamma_Z S_{12} S_{21})^2 + 2S_{11} \Gamma_Z S_{12} S_{21} \cos\theta} + \frac{\Gamma_Z S_{22} \cos(\varphi_Z + \varphi_{22}) - (\Gamma_Z S_{22})^2 + \Gamma_G \Gamma_Z S_{12} S_{21} \cos\psi}{1 + (\Gamma_Z S_{22})^2 - 2\Gamma_Z S_{22} \cos(\varphi_Z + \varphi_{22}) - 2\Gamma_G \Gamma_Z S_{12} S_{21} \cos\psi}; \tag{17}$$

where

$$\varphi_Z + \varphi_{12} + \varphi_{21} - \varphi_{11} = \theta$$
$$\varphi_Z + \varphi_{12} + \varphi_{21} + \varphi_G = \psi$$
$$\varphi_{11} + \varphi_{22} - \varphi_{12} - \varphi_{21} = F$$

Let us write down the conditions for the maximum accuracy of the electrical length control of the transmission path.

1. Based on (14), the condition under which reflections from random irregularities relative to incident waves do not affect the phase $\varphi_{in}$ is $d\varphi_{in}/d\varphi_{11} = 0$. This is provided with

$S_{11} = 0$, but unfeasible on practice, or with $S_{11} = -\Gamma_Z S_{12} S_{21} \cos(\varphi_Z + \varphi_{12} + \varphi_{21} - \varphi_{11})$. Section 4 of this present article examines the influence of microwave discontinuities reflections that randomly distributed along a transmission path.

2. Based on (15), a similar condition for the reflected wave from the load is $d\varphi_{in}/d\varphi_{22} = 0$. The possible solutions are: $\Gamma_Z = 0$, $S_{22} = 0$, $S_{12} = 0$ and $S_{21} = 0$. None of these conditions are met in real systems. Note that the condition:

$$\Gamma_Z S_{12} S_{21} \cos(\varphi_Z + \varphi_{22}) + 2S_{11}\cos\theta - S_{11} \cos F = 0$$

can be approximately reduced to: $\varphi_Z + \varphi_{22} = \frac{\pi}{2} + k\pi$, $k = 1, 2\ldots$

3. From (16) we obtain the condition for the maximum measuring accuracy the electrical length of the path $d\varphi_{in}/d\varphi_{12} = d\varphi_{in}/d\varphi_{21} = 1$. That is possible with $S_{11} = 0$ and $\Gamma_G = 0$.

4. To fully take into account the effect of the load reactance on the output signal phase in the proposed system, it is necessary that $d\varphi_{in}/d\varphi_Z = 2$. It can be seen from expression (17) that this is possible if two conditions are met at the same time:

$$\begin{cases} S_{11} = 0 \\ \cos(\varphi_Z + \varphi_{22}) = \Gamma_Z S_{22} \end{cases}$$

Thus, conditions have been obtained for achieving the maximum measuring accuracy the electrical length of a long microwave path with method of unmodulated reflection from the load. To improve the control accuracy of the electrical length, the most important conditions are:

- eliminate the wave reflection from the signal source, which is achieved by adding a microwave isolator at its output;
- take into account during calibration the influence of microwave discontinuities randomly distributed along the channel on its input reflection coefficient.

## 4. Errors in Measuring the Electrical Length

The use of load reflection to account for random deviation of the electrical length is based on controlling the node shift of the mixed standing and travelling wave. As shown earlier, the accuracy of such control is significantly affected by the presence of random microwave discontinuities along the path. That introduced reflection creates an additional wave shift and therefore introduces a measurement error of the electrical length.

### 4.1. Influence of Microwave Discontinuities a Long Transmission Path on the Node Shift of the Mixed Wave

A long microwave path contains a number of elements that create stray reflections along it, and neither the true parameters of each irregularity, nor their exact location are known in advance. A statistical approach is used to study the effect of microwave discontinuities on the input reflection coefficient of the transmission path.

Each $i$-th random element of the transmission path has a reflection coefficient $\dot{\rho}_i = \rho_i e^{j\varphi_i}$ where $\rho_i$ and $\varphi_i$ are random values of the magnitude and phase of the reflection coefficient, respectively. The random nature of the input reflection coefficient $\dot{\rho}$ is the result of cumulative effects of a large number of reflections from equivalent input irregularities of the path. When assessing the effect of microwave discontinuity on the input reflection coefficient of a path, only the magnitude of its reflection coefficient is essential.

Each discontinuity along the path affects a node shift of the mixed standing and travelling wave $\sin 180° \frac{\Delta l_i}{\lambda} = \rho_i$, where $\Delta l_i$ is the node shift of single irregularity; $\lambda$ is an operating wavelength [20]. Thus,

$$\Delta l_i = \frac{\lambda}{180°} \arcsin\rho_i, \tag{18}$$

where $\lambda$ is a deterministic, and $\rho_i$ is a random quantity having the Rayleigh distribution [28–30]. Since $\Delta l_i$ and $\rho_i$ are linked through the inverse trigonometric function, the distribution law of $\Delta l_i$ cannot be precisely determined. It is necessary to solve the problem using an approximate method. Let us analyze the resulting displacement of the mixed wave $\Delta l$ at the path input. In this case, one can make the assumption that the $\Delta l$ value has a probability distribution close to normal and verify this by constructing a quantile—quantile (q–q) plot.

To obtain statistical data, the magnitude of the reflection coefficient $\rho$ was measured at the long path input—a coaxial cable 20 m long, terminated with mismatched load. The measurements were made at frequencies of 1, 5, and 9 GHz. The conditions for the cable movement along with the turntable rotation were reproduced and 100 samples at each frequency were taken. Histogram plots of the $\arcsin\rho = \frac{\Delta l}{\lambda} 180°$ value for each frequency were constructed (Figure 4a–c). The histogram presents density of $\arcsin\rho$ values (i.e., number of hits $r$ of the values in the corresponding interval; y-axis) over chosen intervals (x-axis).

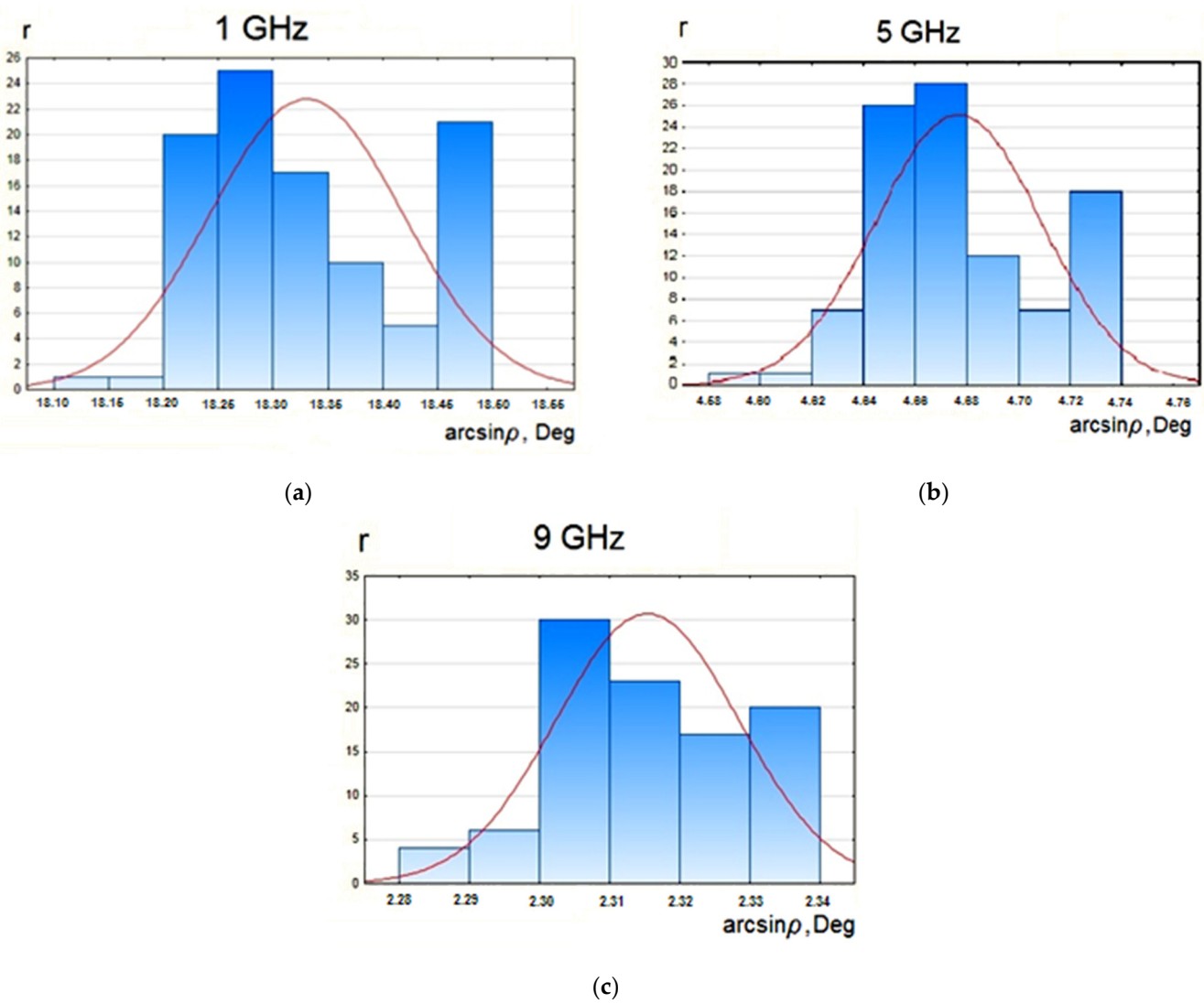

**Figure 4.** Histograms of the mixed wave node shift: (**a**) 1 GHz, (**b**) 5 GHz, (**c**) 9 GHz.

By the type of histogram dependence, one can make an assumption about the distribution law of the investigated quantity. In this case, the histograms in Figure 4a–c allow one to assume a normal law of $\arcsin\rho$.

In order to prove this, it is necessary to construct q-q plots of the random variable arcsinρ. The quantile plot is built on the normal probability paper and presents the investigated quantity arcsinρ (y-axis) over the cumulative frequency (x-axis). The plotted data points (Figure 5a,c) are arranged so that a straight line can be drawn through them.

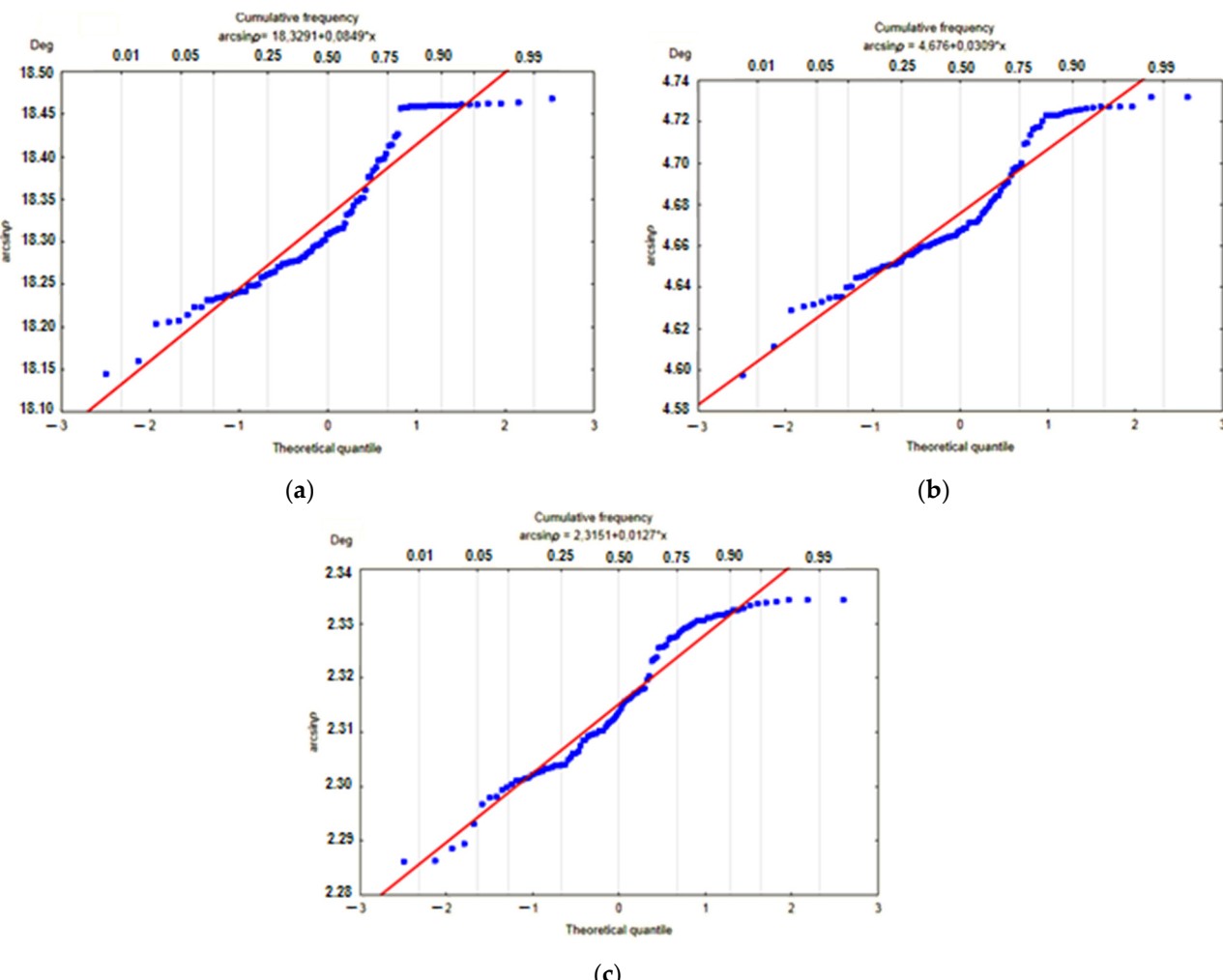

**(a)**

**(b)**

**(c)**

**Figure 5.** Quantile-quantile plots for node shift of the mixed wave: (**a**) 1 GHz, (**b**) 5 GHz, (**c**) 9 GHz.

A straight line drawn on normal probability paper represents a normal distribution. Therefore, considering the deviations of the scatter diagram from a straight line as a result of random influences, we can assume the distribution of arcsinρ is close to normal.

It can be seen that the deviations of the points from the theoretical line are less noticeable for the central frequencies (close to 0.5) than at the edges (close to 0 or 1). That is, when plotting a straight line, which is a theoretical estimate, the end points are less important than the mid-range ones. The q–q plot allows you to determine the main characteristics of the investigated probability distribution.

The mean value on the plot is 0.5; $M1[\text{arcsin}\rho] = 18.33°$ at 1 GHz; $M2[\text{arcsin}\rho] = 4.68°$ at 5 GHz; $M3[\text{arcsin}\rho] = 2.32°$ at 9 GHz. The standard deviation is found as the difference between $M$ and its value at the level of 0.159 along the cumulative frequency (upper x-axis) or $-1$ along the theoretical quantile (lower x-axis). $\sigma_1 = 0.085°$ at 1 GHz; $\sigma_2 = 0.031°$ at 5 GHz; $\sigma_3 = 0.013°$ at 9 GHz.

Thus, the distribution of the electrical length fluctuations, caused by the influence of microwave discontinuities, is close to normal. Accordingly, this makes it possible to correct the accuracy of the electrical length control by the studied method of unmodulated reflection from the load.

### 4.2. Attenuation Influence on Accuracy of the Electrical Length Control of a Inhomogeneous Microwave Path

A statistical study of random microwave discontinuities along the lossless transmission path was carried out in a number of works [31,32]. A key finding of the previous studies is that in a lossless path with irregularities the input wave is determined by equivalent reflection coefficients. As a first approximation, they are the sum of partial reflection coefficients $\rho_i$, translated to the input of the path. In real microwave paths with attenuation losses the magnitude of the incident wave along the path does not remain constant. The further the irregularity is from the beginning of the transmission path, the less influence it makes, which reduces the probable value of the equivalent reflection coefficient. Let us carry out a statistical analysis of the influence of $n$ random irregularities along the real transmission path with attenuation losses on the equivalent input reflection coefficient, taking into account the conclusions obtained.

Let us recalculate the partial reflection coefficients into the common section at the path beginning, taking into account the attenuation losses:

$$\dot{\rho}_i{}' = \rho_i e^{-j2\beta L_i} e^{-2\alpha L_i} e^{j\varphi_i} = \rho_i e^{-2\alpha L_i} e^{-j(2\beta L_i - \varphi_i)} =$$
$$= \rho_i e^{-2\alpha L_i} \cos(2\beta L_i - \varphi_i) + j\rho_i e^{-2\alpha L_i} \sin(2\beta L_i - \varphi_i) = \rho_{xi} + j\rho_{yi},$$

where $\rho_i$ and $\varphi_i$ are the magnitude and phase of the partial reflection coefficient, respectively, $\alpha$ is attenuation of transmission path, $\beta = 2\pi/\lambda$ is phase constant, $L_i$ is the distance to the $i$-th irregularity from the beginning of the path.

Phase shifts $(2\beta L_i - \varphi_i)$ are random variables uniformly distributed over the $[0; 2\pi]$ interval, thus the mean values $\rho_{xi}$ and $\rho_{yi}$ are equal to zero. The input reflection coefficient of the transmission path is defined as the sum of the partial reflection coefficients $\dot{\rho}_i{}'$, translated to the input section of the path.

$$\dot{\rho} = \sum_{i=1}^{n}\left[\rho_i e^{-2\alpha L_i}\cos(2\beta L_i - \varphi_i)\right] + j\sum_{i=1}^{n}\left[\rho_i e^{-2\alpha L_i}\sin(2\beta L_i - \varphi_i)\right] = \rho_x + j\rho_y$$

$\rho_x$ and $\rho_y$ are the sum of normally distributed quantities and, by the central limit theorem, have a normal distribution with probability density $W(\rho_x)$ and $W\left(\rho_y\right)$. Assuming $\rho_x$ and $\rho_y$ to be statistically independent, we find the joint probability as the integral of the product of $W(\rho_x)$ and $W\left(\rho_y\right)$ in polar coordinates $\rho$, $\varphi$.

$$W(\rho) = \int_{0}^{2\pi} W(\rho_x)W\left(\rho_y\right)\rho \, d\varphi = \frac{\rho}{2\pi\sigma^2}\int_{0}^{2\pi} e^{\frac{-\rho_1^2 e^{-2\alpha L_s}}{2\sigma^2}} d\varphi = \frac{\rho}{\sigma^2}e^{-\frac{\rho^2}{2\sigma^2}e^{-2\alpha L}}$$

where

$$\rho_1 = \sum_{i=1}^{n}\rho_i, \qquad L_s = \sum_{i=1}^{n}L_i$$

and $W(\rho_x) = \frac{1}{\sqrt{2\pi}\sigma_x}e^{-\rho_x^2/2\sigma_x^2}$, $W\left(\rho_y\right) = \frac{1}{\sqrt{2\pi}\sigma_y}e^{-\rho_y^2/2\sigma_y^2}$ are probability density distributions with variances $\sigma_x^2 = \sigma_y^2 = \sigma^2 = \frac{1}{2}\sum_{i=1}^{n}\sigma_i^2$.

Thus, the magnitude distribution of the input reflection coefficient is subject to the law of

$$W'(\rho) = \frac{\rho}{\sigma^2}e^{-\frac{\rho^2}{2\sigma^2}}e^{-2\alpha L} \tag{19}$$

Let us show that this is the Rayleigh distribution law. Changing of the parameters $\sigma^2 = A_1$, $e^{-2\alpha L} = A_2$, we equate (19) to some Rayleigh law with parameter $\sigma'$ and variable $\rho'$:

$$\frac{\rho}{A_1}e^{-\frac{\rho^2}{2A_1}A_2} = \frac{\rho'}{\sigma'^2}e^{-\frac{\rho'^2}{2\sigma'^2}}$$

From this equality follows a system of equations:

$$\begin{cases} \frac{\rho}{A_1} = \frac{\rho'}{\sigma'^2} \\ \frac{\rho^2}{2A_1}A_2 = \frac{\rho'^2}{2\sigma'^2} \end{cases} \tag{20}$$

Solving the system (20), we obtain:

$$\begin{cases} \rho' = \rho e^{-2\alpha L} \\ \sigma'^2 = \sigma^2 e^{-2\alpha L} \end{cases}$$

Thus, the desired magnitude distribution of the input reflection coefficient is subject to the law of Raleigh.

$$W(\rho') = \frac{\rho'}{\sigma'^2}e^{-\frac{\rho'^2}{2\sigma'^2}}$$

Its parameter is the most probable value of the input reflection coefficient $\rho'_m = \sigma' = \sigma\sqrt{e^{-2\alpha L}}$, corresponding to the top of distribution. That is, the input reflection coefficient variance $\sigma'^2$ of the transmission path with attenuation losses is $e^{2\alpha L}$ times less than in the case of a lossless path $\sigma^2$, by reducing the additional node shift $\Delta l$ of the wave.

Let us find the difference between node shifts for a transmission path with and without attenuation losses. The magnitude of the input reflection coefficient is assumed to be $\rho'_{0.9} = 2.14\rho'_m = 2.14\sigma\sqrt{e^{-2\alpha L}}$, where $\rho'_{0.9}$ is reflection coefficient with level of probability $P(\rho < \rho'_{0.9}) = 0.9$. Relative node shift of the wave:

$$\Delta l'/\lambda = \frac{1}{180°}\arcsin\rho'_{0.9} = \frac{1}{180°}\arcsin\left(\rho_{0.9}\sqrt{e^{-2\alpha L}}\right)$$

where $\rho_{0.9} = 2.14\sigma$.

The difference in relative shifts in terms of the electrical length for a transmission path with and without attenuation losses:

$$\delta\beta l = \Delta\beta l - \Delta\beta l' = 2\left(\arcsin\rho_{0.9} - \arcsin\left(\rho_{0.9}\sqrt{e^{-2\alpha L}}\right)\right)$$

Figure 6 depicts the dependence $\delta\beta l(2\alpha L)$. As the attenuation over transmission path increases, the node shift of the wave decreases, and consequently the shift difference grows.

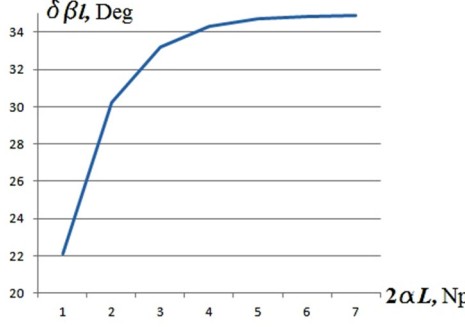

**Figure 6.** Note shift difference of the mixed standing and travelling wave.

As an example, for $2\alpha L = 4.6$ Np, and $\rho_{0.9} = 0.3$, the node shift in the lossless path is equal to $35°$, whereas in the path with attenuation it is $3.4°$. The standing wave arises due to reflection from microwave discontinuities along the transmission path. Thus, an increase in the attenuation losses of the path reduces the probable node shift of the mentioned standing

wave, and, therefore, reduces their influence on the result of antenna measurements. However, with a large attenuation, the standing wave ratio of the transmission path decreases, and the standing wave mode is observed near the AUT only. This makes it difficult to control the electrical length using the main load reflection. In addition, with increasing path attenuation, less power is supplied to the antenna, which reduces the signal-to-noise ratio during measurements. Optimization of antenna measurements and measurement conditions may be the subject of separate studies.

## 5. Measurement Results

Over the past decades, meters of the scattering parameters, also known as network analyzers, have become widespread in the analysis of microwave devices. A number of different calibration types of vector network analyzers (VNA) and their variations are known today [33–36]. The calibration process of VNA is designed to minimize the systematic term of the measurement uncertainty. The extension of this tool makes it possible to implement in practice the method proposed for taking into account random measurement errors in the moving long microwave paths. In the general case, the considered algorithm for correcting the measurement results is applicable to any family of calibration types. The following describes the main steps in the process regarding SOLT calibration.

### 5.1. 1P Calibration

If it is necessary to measure the reflection coefficient at the input of a device under test (DUT), a 3-term error model of a one-port (1P) VNA is considered. This model is derived from a 12-parameter model of a two-port (2P) VNA (EURAMET CG No. 12 «Guidelines on the Evaluation of Vector Network Analysers (VNA)» Version 3.0) through simplification. To calibrate the VNA, it is necessary to measure the reflection from three different known loads—open circuit (OC), short circuit (SC) and matched load (ML). As a result of solving a system of linear equations, the following VNA error factors are determined: directivity $\dot{E}_D$, source match $\dot{E}_S$, and reflection tracking $\dot{E}_R$. Then, the corresponding correction for the reflection coefficient of the measured load is written as:

$$\dot{\Gamma}^a = \frac{\dot{\Gamma}^m - \dot{E}_D}{\dot{\Gamma}^m \dot{E}_S - \dot{E}_D \dot{E}_S + \dot{E}_R} \tag{21}$$

where $\dot{\Gamma}$ is the reflection coefficient of the DUT; subscripts "*a*" and "*m*" refer to actual and measured value, respectively; $\dot{E}$—error terms of 1P VNA model.

When moving from the classic 1P calibration, which assumes static measurements, i.e., fixed cables, to the case of dynamic process, no additional measuring equipment is required. It is assumed that each error term now depends not only on frequency, but also on the spatial coordinates of the measuring stand components—measuring antenna and antenna under test. In other words, the dynamic calibration allows one to determine terms of the VNA error model and perform an unique correction of the measured data depending on spatial coordinates of antennas. Since this technique implies data post-processing, the measurements associated with it can be carried out both before and after the main measurements of the DUT. The process for measuring the reflection of calibration standards should be the same as for DUT measuring, that is, process is repeated for the available OC, SC and ML standards, connected consistently to the measuring cable instead of the DUT.

For example, the considered calibration can be applied in the far field antenna measurements jointly with the transmission coefficient normalization when measuring a reference antenna. The described process takes advantage of the unmodulated load reflection method and assumes ensuring the repeatability of the measurement conditions.

### 5.2. 2P Calibration

The two-port VNA model contains 12 error terms. In addition to the mentioned 1P VNA terms, the following errors are included in the model: load match $\dot{E}_L$, transmission

tracking $\dot{E}_T$ and isolation $\dot{E}_X$. Due to the VNA model symmetry, the obtained six error terms describe it in one of the probing directions, e.g., forward. Similar six parameters describe the error model in the other—reverse—probing direction. Conventional 2P error correction is carried out in three stages: (a) 1P calibration of ports; (b) response measurement of two ports through the adapter; (c) optional measurement of error terms $\dot{E}_X$ with two MLs, otherwise considered $\dot{E}_X = 0$.

Proposed dynamic 2P error correction assumes the use of a phase-stable test cable as a thru adapter. During the measurement, the test cable is subjected to mechanical impact, thus the thru adapter will also be affected in the same way. It is allowed to use an assembly of measuring cables, provided that the phase-stable unit is subjected to mechanical impact as shown in Figure 7a. The proposed extension adapter consists of one or two phase-stable cables at the ends and a conventional measuring cable to ensure the required length between ports. The scattering parameters of such an adapter must be known, i.e., measured separately. The described approach is similar to calibration with a known adapter («Defined Thru») (AN 1287-11 «Specifying Calibration Standards and Kits for Agilent vector Network Analyzers», Agilent Tech., USA, 2009), but differs in that the calibrated cables are not static during the measurement.

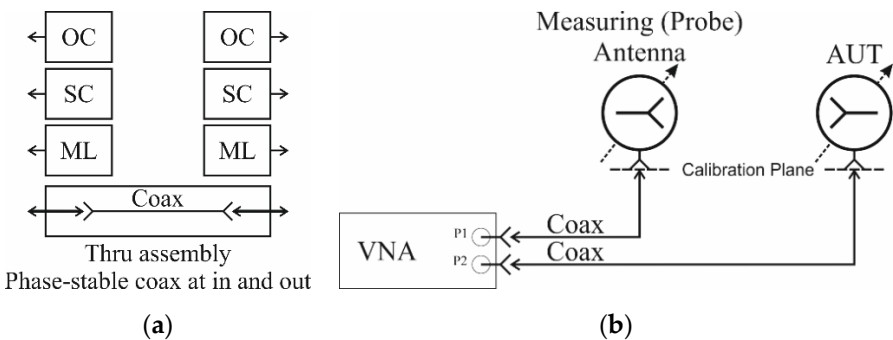

(**a**)   (**b**)

**Figure 7.** Dynamic calibration: (**a**) full two-port SOLT calibration standards with defined thru; (**b**) generalized antenna measurement diagram with 2P VNA.

A generalized scheme of the measuring setup is shown in Figure 7b, where the dotted line at the antennas indicates possible changes in spatial coordinates during measurement and the dashed one indicates the calibration plane.

Similar to the conventional 2P calibration, the first step is to consistently measure the reflection of standards in calibration plane, where the measuring antenna and the AUT are connected, respectively. The second step is to measure response between ports through the adapter. Further, the isolation between ports is optionally measured to determine the $\dot{E}_X$ parameter.

The technique was tested at the far-field test setup. The scheme of the measuring stand is shown in Figure 8. The horn antenna of the measuring complex was taken as AUT. Measurements were carried out in the frequency range of (0.8–6) GHz. Figure 9 shows the corrected magnitude $\Gamma_{in}$ and phase $\varphi_{in}$ components of the AUT reflection coefficient for azimuth positions of the turntable. The gray solid line depicts direct measurement and the red one after the data correction has been applied. The stability of the corrected parameter along the azimuth coordinate indicates the measurements repeatability. The obtained reflection coefficient coincides with the measurements corrected via built-in VNA calibration process and is in good agreement with the antenna manufacturer's specifications.

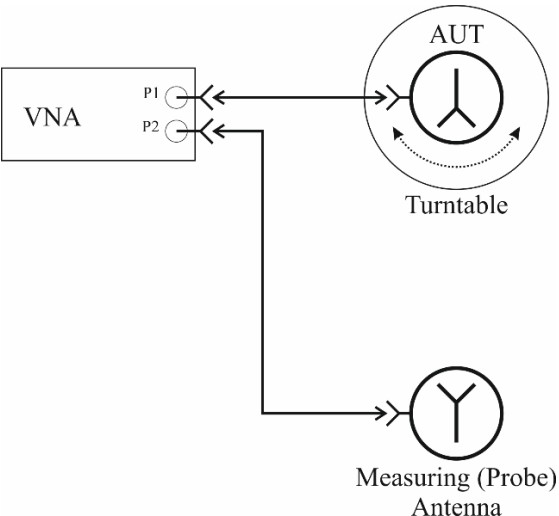

**Figure 8.** Far field antenna measurement diagram.

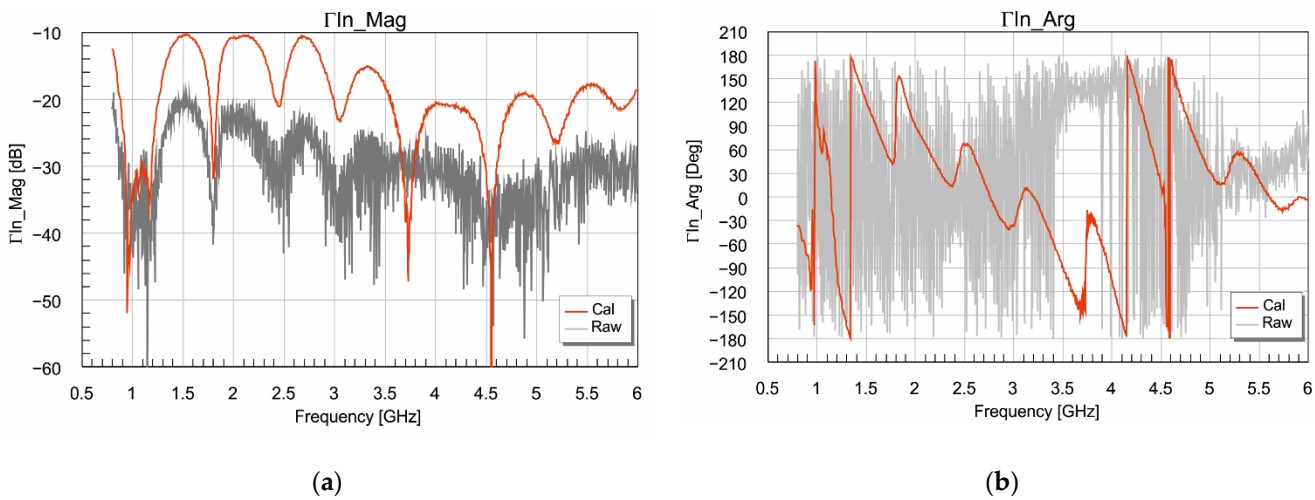

(**a**) (**b**)

**Figure 9.** Reflection coefficient of the AUT: (**a**) magnitude and (**b**) phase.

Figure 10 shows an example of the experimental evaluation of the measuring cable contribution depending on the angle of the AUT rotation at given frequency. The purple and blue lines correspond to the co-polarized and cx-polarized radiation patterns of the AUT. The red line corresponds to the total radiation pattern. Black lines correspond to radiation patterns after dynamic calibration is applied. And the teal line corresponds to the difference in the total characteristics obtained.

Figure 11 shows a comparison of the traditional 2P VNA calibration and the presented dynamic by difference in the AUT radiation patterns obtained. The first curve (purple line) corresponds to the use of error correction during traditional full two-port SOLT VNA calibration in the specified position of the measuring stand. That is, for any other positions of the stand, the error terms of the VNA model are assumed to be the same. The curve 2 (teal line) corresponds to the case of dynamic calibration, when calibration process is carried out in each position of the measuring stand. Curve 3 (blue line) shows the AUT RP difference when applying normal and dynamic calibrations.

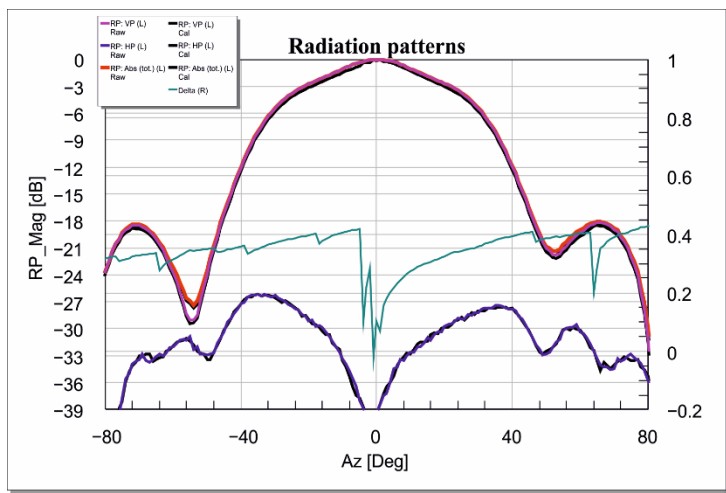

**Figure 10.** AUT radiation patterns at frequency f = 3.75 GHz.

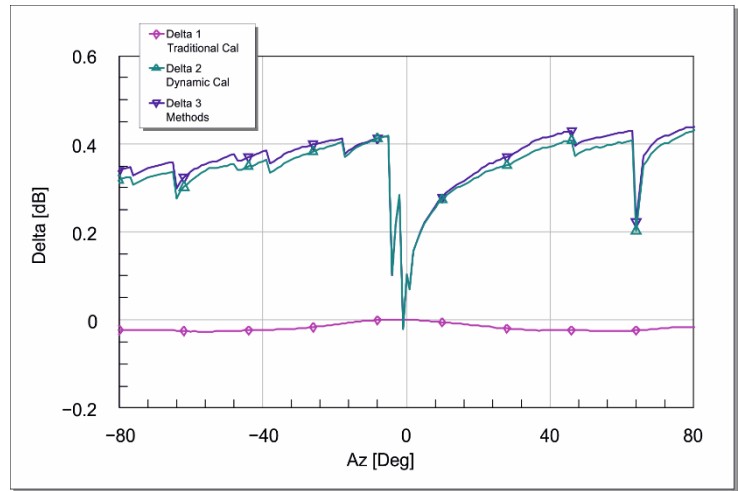

**Figure 11.** Difference of AUT radiation patterns for traditional (static) and dynamic calibrations at frequency f = 3.75 GHz.

Measurement of the radiation pattern was carried out in the usual way for the main and cross-polarization. The obtained error in the magnitude of the transmission coefficient is on average 0.26 dB over studied frequency band and reaches a peak value of 0.58 dB.

## 6. Conclusions

In this paper, mathematical models of long microwave paths with and without random errors correction are proposed and analyzed. It is shown that it is possible to minimize the total measurement error using dynamic calibration. The authors carried out a study of the electrical length control method of a long microwave path based on unmodulated reflection from the load. A mathematical model of a long microwave path has been developed. The sensitivity matrix elements of the system under study were defined. The dependencies of the elements of the system state vector on the system parameters were constructed.

Using theory of sensitivity, the mathematical expressions of conditions were obtained to achieve maximum accuracy of measuring the electrical length of a long microwave path. The usefulness of this method for antenna measurements is shown. Relationships have been found to assess the effect of microwave discontinuities on the control accuracy through a statistical approach.

The authors propose some improvements to the procedure for error correction in the antenna measurements. The possibility of dynamic error correction of antenna mea-

surements with a long test path under systematic mechanical impact on it has been experimentally proved. In turn, this reduces the contribution of systematic error in the uncertainty budget. The presented technique can also be considered as a diagnosis tool for measurement cables of the antenna test setup and used in various microwave signal routing units.

**Author Contributions:** Conceptualization, E.D. and M.S.; methodology, E.D.; software, M.S.; validation, M.S.; formal analysis, E.D.; investigation, M.S.; resources, M.S.; data curation, E.D.; writing—original draft preparation, E.D. and M.S.; writing—review and editing, E.D.; visualization, M.S.; supervision, M.S.; project administration, E.D.; funding acquisition, E.D. All authors have read and agreed to the published version of the manuscript.

**Funding:** This research was funded by state assignment of the Ministry of Science and Higher Education of the Russian Federation, project No. FSFF-2020-0015.

**Institutional Review Board Statement:** Not applicable.

**Informed Consent Statement:** Not applicable.

**Data Availability Statement:** Not applicable.

**Conflicts of Interest:** The authors declare no conflict of interest.

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
