# Peer review of "Dynamic Correction of the Influence of Long Measuring Path Irregularity in Antenna Tests"

_applsci, doi:10.3390/app11178183_

Round 1
Reviewer 1 Report
The authors have presented a good study on "Dynamic correction of the influence of long measuring path 2 irregularity in antenna tests" with the aim to develop a theoretical background and perform experimental verification for taking into account the effect of random microwave discontinuities on the transmission path characteristics.
Although the results were sound technical, I have some concerns that should be revised carefully and resubmit the article for further assessment.
1: The article should be revised carefully, It is highly recommended to send your article to some native speaker for the removal of grammatical errors. Here are some examples:
a) In line no. 8, "influence" should be replaced with "the influence"
b) In line no. 11, "measurements" should be replaced with "the measurements"
c) In line no. 17, "path's" should be replaced with "the path's"
d) In line no. 19, "theory" should be replaced with "the theory"
e) In line no. 21, "off" should be replaced with "in"
f) In line no. 30, The sentence should not be started with "Or"
2: Throughout the manuscript, the authors used very long sentences which are not prefered by the reader. Thus to grab the reader attention small yet brief sentences should be used. Instead of using a lengthy sentence of 30-40 words, the author should use multiple sentences of 10-15 words. Here is an example of a lengthy sentence:
"Or monopulse antenna arrays, which have a high value of difference pattern null slope, and therefore a high resolution, that expand the accessible amount of information about the propagation of electromagnetic waves from sources of radiation or reflection in the space around."
3: In the Introduction, paragraphs no. 1 and 2. no proper reference was cited for various terms like active 29 electronically scanned arrays (AESA). Are the authors introduced them?
4: Some states of the artwork on the antenna should be cited at the proper place in the introduction section. Here are few examples
a) Ghaffar, Adnan, et al. "Design and Realization of a Frequency Reconfigurable Multimode Antenna for ISM, 5G-Sub-6-GHz, and S-Band Applications." Applied Sciences 11.4 (2021): 1635.
b) Zahra, H., et al. "GHz Broadband Helical Inspired End-Fire Antenna and Its MIMO Configuration for 5G Pattern Diversity Applications. Electronics 2021, 10, 405." (2021).
4: The quality of Figures, especially fig. 4 should be improved for readability.
5: It is highly recommended to use some graph tool to plot various graphs to improve the presentation of the paper.
6: There is a lot of formatting errors found throughout the manuscript. Here are few examples:
a) After all figure's caption space should be given from the text.
b) The size of the figures should be kept the same for all figures.
c) Conclusion should not be written in bullets. It is highly recommended to summarize the conclusion briefly.
Reviewer 2 Report
I could not follow some of the derivations. Please verify them.
The use of one-port and two-port calibration of VNA's is very well known. After all of the derivations, it appears you use the standard calibration methods. What is novel about your method?

Round 2
Reviewer 1 Report
Authors carefully addressed all concerns.
Reviewer 2 Report
There are some parts that still need clarification.
- Page 4, line 150, you state "it is possible to control the electrical length." Yes, but only if you adjust the length of the cables or add a matching circuit. In the sentences above you are talking about measuring the change in the electrical length by looking at the standing wave pattern. Explain how you adjust the electrical length.
- On page 13, near the top, you claim rightly that a cable or system with high losses is less susceptible to path discontinuities. I do not think you need a statement that this requires further research. It is fairly well known.
- To me, it is still not clear how the calibration is being performed. Are you doing a calibration at every angle of rotation? If you are not, then you are using a very standard calibration method. To prove the method I think you are proposing is better than the single, static, calibration, I suggest you compare your dynamic calibration to a static, one angle, calibration. Comparing to raw data does not prove your method is better.
Round 3
Reviewer 2 Report
Thank you for making the revisions. The paper is much better now.